# SnapKV: LLM Knows What You Are Looking for before Generation

**Yuhong Li**[1][*]   **Yingbing Huang**[1][*]   **Bowen Yang**[2]   **Bharat Venkitesh**[2]   **Acyr Locatelli**[2]
**Hanchen Ye**[1]   **Tianle Cai**[3]   **Patrick Lewis**[2]   **Deming Chen**[1]
[1] University of Illinois Urbana-Champaign   [2] Cohere   [3] Princeton University
[1]{leeyh, yh21, hanchen8, dchen}@illinois.edu
[2]{bowen, bharat, acyr, patrick}@cohere.com   [3]tianle.cai@princeton.edu

## Abstract

Large Language Models (LLMs) have made remarkable progress in processing extensive contexts, with the Key-Value (KV) cache playing a vital role in enhancing their performance. However, the growth of the KV cache in response to increasing input length poses challenges to memory and time efficiency. To address this problem, this paper introduces SnapKV, an innovative and fine-tuning-free approach that efficiently minimizes KV cache size while still delivering comparable accuracy in real-world applications.

We discover that each attention head in the model consistently focuses on specific prompt attention features during generation. Meanwhile, this robust pattern can be obtained from an 'observation' window located at the end of the prompts. Drawing on this insight, SnapKV automatically compresses KV caches by selecting clustered important KV positions for each attention head. Our approach significantly reduces the growing computational overhead and memory footprint when processing long input sequences. Specifically, SnapKV achieves a consistent decoding speed with a 3.6x increase in generation speed and an 8.2x enhancement in memory efficiency compared to the baseline when processing inputs of 16K tokens. At the same time, it maintains comparable performance to the baseline models across 16 long sequence datasets. Moreover, SnapKV can process up to 380K context tokens on a single A100-80GB GPU using HuggingFace implementation with minor changes, exhibiting only a negligible accuracy drop in the Needle-in-a-Haystack test. Further comprehensive studies suggest SnapKV's potential for practical applications. Our code is available at https://github.com/FasterDecoding/SnapKV.

## 1   Introduction

Many leading LLMs have started to handle longer contexts, overcoming the difficulties in context maintenance and attention mechanism scalability, such as GPT-4 [1] and Command-R [2] with context length 128K, Claude-3 [3] with 200K, and Gemini-Pro-1.5 with 1M [4]. Despite their impressive capabilities, LLMs still face significant challenges when dealing with long context prompts. Specifically, the KV cache in attention calculation becomes less efficient when processing long context. During inference time, as prompt length increases, the decoding latency per step grows linearly due to the attention calculation across past KVs. Moreover, the large KV cache requires significant memory capacity, increasing hardware demands and limiting model scalability.

---

[*]equal contribution

38th Conference on Neural Information Processing Systems (NeurIPS 2024).

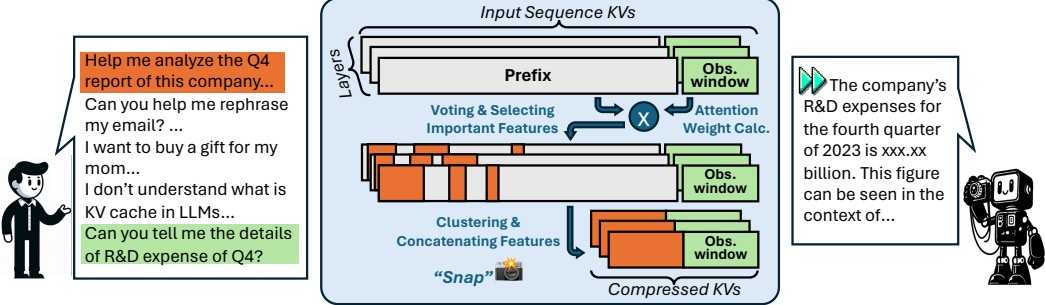

Figure 1: The graph shows the simplified workflow of SnapKV, where the orange area represents the cluster of features per head selected by SnapKV. These features are then used to form new Key-Value pairs concatenated with the features in the observation window. Together, the selected prefix and observation windows constitute the new KV cache utilized for the generation.

There are many approaches to mitigate these problems, such as KV cache eviction during generation stage [5–8]. However, most of these methods lack a detailed evaluation in long-context settings. Moreover, they mainly focus on compressing the KV cache appended during decoding steps, while overlooking the realistic problem of compressing KV cache for prompts, which is typically the bottleneck in memory efficiency. In practical applications like chatbots and agents, where prompts range from multi-turn conversations to extensive articles or codebases [1, 9–11], prompts are often much larger than generated responses such as summaries and code pieces, thus creating significant inference latency and memory utilization overhead. Additional challenge lies in compressing KV cache for such vast prompts without losing crucial information for accurate generation, especially in scenarios with various noisy contexts.

In our paper, we find an vital attention allocation phenomenon: only a subset of prompt tokens convey essential information for response generation, and these tokens remain unchanged during generation. To validate the robustness, we design extensive experiments across diverse prompts in terms of length, format, and content. From our observations, we derive an innovative and intuitive method, SnapKV, which can smartly identify the attention allocation pattern and compress the KV cache for long sequence prompts without compromising the model's accuracy. With its comprehensive design, SnapKV demonstrates its effectiveness on various datasets and can be easily integrated into popular deep-learning frameworks with just a few code adjustments. Our contributions are as follows:

- We design experiments to explore the attention allocation pattern during generation, focusing on two key questions:
  1. Is there a consistent attention allocation pattern for input sequence tokens?
  2. Is it feasible to identify this pattern prior to the generation stage?

  Our finding suggests that for LLMs, the attention allocation of most input sequence tokens stay consistent during generation. Thus, *LLMs knows what you are looking for before generation.*

- Inspired by our observations above, we develop an efficient and fine-tuning-free algorithm, SnapKV, which efficiently identifies critical attention features and compresses KV cache correspondingly with minimal model modification (See Fig. 1).

- We evaluate SnapKV across diverse LLMs and long-sequence datasets. SnapKV shows comparable accuracy with full KV caching method while achieving improved decoding speed and memory efficiency. Meanwhile, we conduct the pressure test with Needle-in-a-Haystack to further demonstrate its memory efficiency and information retrieval ability.

## 2 Related Works

Many previous works compress the KV cache by selectively dropping KVs using different algorithms. In StreamLLM [5], only the most recent tokens and attention sinks (first few tokens) are retained

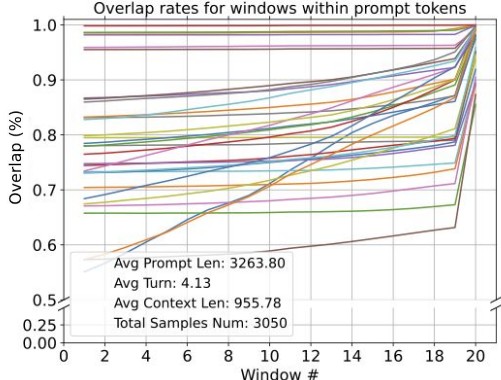
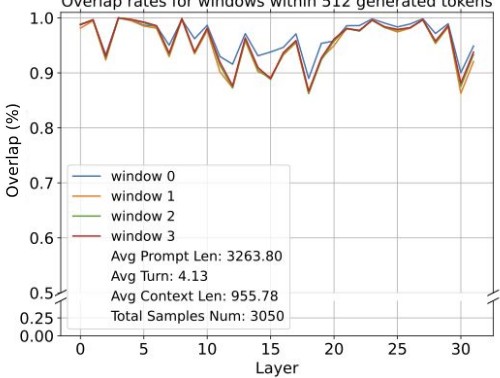

Figure 2: The overlap rates between attention features of the input sequence, selected by various windows along the input and during generation, with each line representing a model layer.

Figure 3: The layer-wise overlap rates between input sequence attention features selected by the last window of input sequence and those selected by 4 windows along generation.

to reduce the KV cache size, making it lose the important information carried by the discarded middle tokens [2]. Heavy-Hitter Oracle (H2O) [6] introduces a policy that greedily drops KVs during generation based on a scoring function derived from cumulative attention. While this approach effectively compresses the KVs appended to the cache during generation, it overlooks compression of prompt KVs, which is crucial for reducing memory and computational overhead. Building on a similar concept, Adaptive KV Compression (FastGen) [8] implements a dual-phase algorithm that encompasses four KV cache compression policies. Initially, it identifies optimal policies through profiling results obtained from prompt encoding. Subsequently, it dynamically evicts caches during the generation phase based on these policies. Nonetheless, it faces the similar problem with H2O. ScissorHands [7] focuses on identifying and retaining pivotal tokens that exhibit a consistent attention weight pattern with previous token windows during generation steps. However, this method concentrates solely on the window of previous pivotal tokens in generation and neglects the extensive prompt that contains essential information for generating accurate responses. This oversight could lead to an inability to extract detailed information from prompts.

In summary, existing methods have not effectively addressed the challenges encountered in real-world applications, where prompts are exceptionally long yet require accurate information retrieval. Although these techniques may reduce the KV cache size during generation, they do not address the primary challenges of understanding complex prompt contexts, leaving critical issues unresolved.

## 3 Observations

In this section, we present our observations regarding the attention allocation patterns in the Query-Key matrix during token generation. Our analysis utilizes samples from Ultrachat [12], a multi-turns, high-quality instruction dataset consisting of 1.4 million dialogues. We further filter the sequences with response length greater than 512 and prompt length greater than 3k. Our findings are concluded into two key observations as follows:

- **Pattern can be identified before generation.** In this experiment, we split the attention features of input sequence of each layer into multiple windows, each with 128 tokens, and calculate the averaged attention weights of the last 20 windows separately. To understand the attention allocation patterns along input sequences, we calculate the overlap rates between *important* attention features of input sequence (those with high average attention weights) identified by each window and the actual ones used by generation. The experimental results are shown in Fig. 2.

---
[2]https://github.com/mit-han-lab/streaming-llm?tab=readme-ov-file#faq

We observe that the last window of input sequence recognizes highly similar attention allocation pattern with the actual generation.

- **Pattern is consistent during generation.** We study if the positions of features identified as crucial in the last window of input sequence maintain their significance in the subsequent token generation. In the experiment, we split the generated tokens into 4 windows for every layer, each spanning 128 tokens, to compute the averaged overlap rates of these windows versus the last window of input sequence. As shown in Fig. 3, active attention features of input sequence obtained from the last window exhibit remarkable consistency throughout the generation process, as evidenced by high overlap rates.

## 4 SnapKV

In the attention mechanism, the growth in prompts will significantly increase time complexity for generation due to the Query-Key matrix multiplication. `SnapKV` addresses this issue by maintaining a constant amount of prompt KVs during generation, significantly reducing serving times for long-context LLMs. To structure our method coherently, we propose the following terminologies:

- **Prompt Length** ($L_{\text{prompt}}$)**:** The total length of the user-provided input.

- **Observation Window** ($L_{\text{obs}}$)**:** The last segment of the prompt. This window is crucial for analyzing the influence of different contexts on attention allocation patterns.

- **Prefix Length** ($L_{\text{prefix}}$)**:** The length of the input preceding the observation window. It is part of the prompt and does not include the observation window. Overall, we have:

$$L_{\text{prompt}} = L_{\text{prefix}} + L_{\text{obs}} \tag{1}$$

- **Voting:** The process of calculating attention weights for each query within the observation window across all heads, aggregating these weights to highlight the prefix positions that are considered most significant. For a single batch of sequence, formally:

$$\mathbf{C} = \sum_{i=0}^{L_{\text{obs}}} \mathbf{W}_{\text{obs}}[:, i, :] \tag{2}$$

$$I = \text{Top}_k(\mathbf{C}, k) \tag{3}$$

where $\text{Top}_k(\mathbf{C}, k)$ selects the indices $I$ of the top $k$ values in tensor $\mathbf{C}$ per head. $k$ is defined as $\lfloor (1 - p) \times L_{\text{prefix}} \rfloor$, where $p$ stands for the compression rate. The tensor $\mathbf{W}_{\text{obs}} \in \mathbb{R}^{N \times L_{\text{obs}} \times L_{\text{prefix}}}$ represents the subset of the prompt softmax-normalized attention features over $N$ heads.

- **Hit Rate:** We define attention features above a predefined threshold $\theta$ during generation as *important* features. The hit rate, $H$, is the number of important features successfully selected by the previous voting process over the total number of important features. $H$ quantifies the effectiveness of the voting mechanism and is calculated as follows:

$$\mathbf{M}_{\text{vote\_obs}} = \text{zeros\_like}(\mathbf{A}_{\text{cur}}) \tag{4}$$

$$\mathbf{M}_{\text{vote\_obs}}[I] = 1 \tag{5}$$

$$\mathbf{M}_{\text{threshold\_cur}} = \mathbb{1}(\mathbf{A}_{\text{cur}} > \theta) \tag{6}$$

$$\mathbf{O} = \mathbf{M}_{\text{threshold\_cur}} \wedge \mathbf{M}_{\text{vote\_obs}} \tag{7}$$

$$H = \frac{\sum \mathbf{O}}{\sum \mathbf{M}_{\text{threshold\_cur}}} \tag{8}$$

$\mathbf{A}_{\text{cur}} \in \mathbb{R}^{N \times L_{\text{prefix}}}$ represents the attention features between the current generated query and prefix keys. $\mathbf{M}$ selects attention features by indices. The threshold operation filters $\mathbf{A}_{\text{cur}}$ to retain only features with values over $\theta$, indicating important attention activations. The $\mathbf{O}$ measures the overlap between attention features selected by $\mathbf{M}_{\text{threshold\_cur}}$ and $\mathbf{M}_{\text{vote\_obs}}$, quantifying the alignment of the current attention with previously identified important features. The hit rate $H$ is then computed as the ratio of the sum of overlap $\mathbf{O}$ to the sum of important features $\mathbf{M}_{\text{threshold\_cur}}$, providing a metric for the efficacy of the attention mechanism in recognizing and emphasizing important attention features within the context. We use $\mathcal{H}(\mathbf{M}_{\text{threshold\_cur}}, \mathbf{M}_{\text{vote\_obs}})$ to denote combination of Eq. 7 and Eq. 8.

## 4.1 Observation Window-based Algorithm

The core approach of `SnapKV` involves identifying and selecting the most crucial attention features per head to create the compressed KV cache. Listing 1 shows the PyTorch-style pseudo code of `SnapKV`. Overall, `SnapKV` operates through two stages as follows:

- **Vote for important previous features.** By the voting process defined above (Eq. 2), we select the important attention features based on the observation window. Sec. 3 highlights the consistency of the attention allocation pattern within observation windows throughout the generation, suggesting that these selected attention features are also vital for subsequent generation. Furthermore, we implement clustering to retain the features surrounding the selected attention features (Sec. 4.3). Line 8-17 shows the pseudo code of the voting process.

- **Update and store compressed keys and values.** We concatenate the selected attention features with all features within the observation window, which encompasses all features containing the necessary prompt information. Line 18- 24 shows the compressing process. The concatenated KVs are stored for later use in generation, thereby saving memory usage.

```python
def snap_kv(query_states, key_states, value_states, window_size, max_capacity_prompt,
    kernel_size):
    bsz, num_heads, q_len, head_dim = query_states.shape
    # Ensure it is the prompt phase.
    assert key_states.shape[-2] == query_states.shape[-2]
    if q_len < max_capacity_prompt:
        return key_states, value_states
    else:
        # Compute attention weights of observing window's queries and prefix context's Keys.
        attn_weights = compute_attn(query_states[..., -window_size:, :], key_states,
    attention_mask)
        # Sum the weight along the query dimension.
        vote = attn_weights[..., -window_size:, :-window_size].sum(dim=-2)
        # Apply 1D pooling for clustering.
        pool_vote = pool1d(vote, kernel_size=kernel_size, padding=kernel_size//2, stride=1)
        # Select top-k indices based on the pooled weights to identify important positions.
        indices = pool_vote.topk(max_capacity_prompt - window_size, dim=-1).indices
        # Expand the indices to match the head dimension for gathering.
        indices = indices.unsqueeze(-1).expand(-1, -1, -1, head_dim)
        # Gather the compressed past key and value states based on the selected indices.
        k_past_compress = key_states[..., :-window_size, :].gather(dim=2, index=indices)
        v_past_compress = value_states[..., :-window_size, :].gather(dim=2, index=indices)
        k_obs = key_states[..., -window_size:, :]
        v_obs = value_states[..., -window_size:, :]
        key_states = torch.cat([k_past_compress, k_obs], dim=2)
        value_states = torch.cat([v_past_compress, v_obs], dim=2)
        return key_states, value_states
```

Listing 1: Implementation of `SnapKV` in pseudo PyTorch style.

## 4.2 Robustness Analysis of Hit Rate

To understand the robustness of the observation window-based algorithm, we analyze its hit rate on multiple long documents QA datasets including QMSum [13], a query-based multi-domain meeting summarization; Openreview [14], a collection of papers from `openreview.net`; SPACE [15], an extractive opinion summarization in quantized transformer spaces. The model we probe is `Mistral-7B-Instruct-v0.2`. Overall, we want to answer the following two questions:

1. Does the nature of instructions in the prompt affect the hit rate?
2. Does the context and instruction positioning affect the hit rate?

### 4.2.1 Contextual Dependency of Patterns

We analyze whether instructions will affect the selection of important features even if the provided context is the same. Our experiment utilizes different instructions on the same document and selects the important features based on the observation window that consists of both the instructions and their corresponding responses. Then we calculate the hit rates between important features selected by different instruction-response pairs $(A, B)$ within the same document by using $\mathcal{H}(\mathrm{M_{vote\_A}}, \mathrm{M_{vote\_B}})$

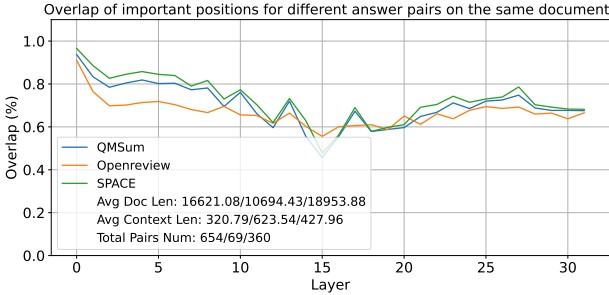

Figure 4: The layer-wise overlap of important positions utilized by different question-answer pairs in the same dataset.

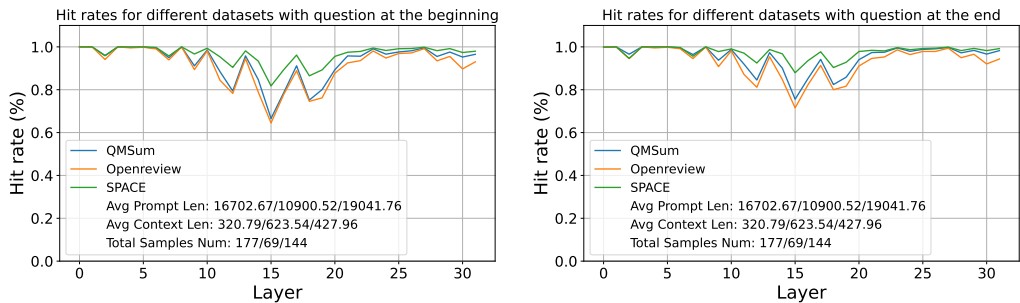

Figure 5: The layer-wise average hit rate of important positions used by prompts with questions at the beginning and the end.

as previously defined. By varying the instructions, we observe that different instructions prioritize different prefix attention features, as indicated by the descending trend in hit rates shown in Fig. 4. Our findings reveal an interesting aspect of KV cache in LLMs: the important attention features change with different instructions. This variability challenges the effectiveness of static compression methods that depend on constant weighted importance or fixed policies [7, 6, 8]. Thus, the complex relationship between context and related KV cache emphasizes the need for context-aware compression strategies and highlights the capability of `SnapKV` that recognizes this dynamic. In contrast, context-independent compression fail in capturing the dynamic, resulting in a misalignment between the attention distribution during profiling and inference, diminishing the generation quality of LLMs.

### 4.2.2 Invariance to Instruction Positions

Our analysis also extends to the significance of instruction positioning on the interpretability of LLMs and their selection of important features. We calculate the average hit rate for the responses using the same observation window size as in the previous experiment. Our results shown in Fig. 5 indicate that across all three datasets, the hit rates are consistently high regardless of whether instructions are positioned before or after extensive supplementary contexts. This consistency suggests that `SnapKV` is able to identify attention allocation patterns regardless of the question's positions.

### 4.3 Efficient Clustering via Pooling

In LLMs, information retrieval and generation rely on features with high attention weight and are supplemented by copying the rest of features in context using induction heads [16]. Hence, naively selecting the top features results in retaining only portions of details and then losing the completeness of the information. For example, such compression might cause the LLMs to retrieve only the country code of a phone number and hallucinate the rest. Our experiment also revealed that only selecting the features with the highest weights is insufficient (Sec. 5.2). Such sparse selection risks compromising

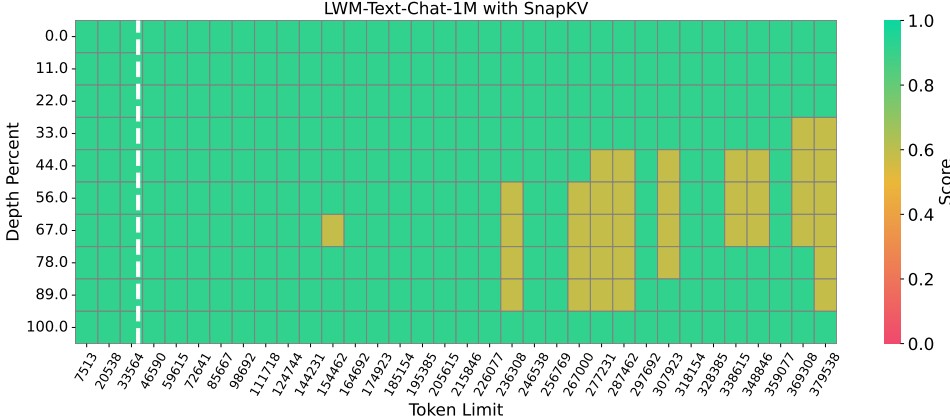

Figure 6: Needle-in-a-Haystack test performance comparison on single A100-80GB GPU, native HuggingFace implementation with only a few lines of code changed. The x-axis denotes the length of the document (the "haystack") from 1K to 380K tokens; the y-axis indicates the position that the "needle" (a short sentence) is located within the document. For example, 50% indicates that the needle is placed in the middle of the document. Here LWMChat with `SnapKV` is able to retrieve the needle correctly before 140k and with only a little accuracy drop after. Meanwhile, the original implementation encounters OOM error with 33k input tokens (white dashed line).

the contextual integrity encapsulated in between features, thereby reducing accuracy. Based on the insights, we propose a fine-grained clustering algorithm utilizing a pooling layer shown in Line 13.

# 5   Experiments

In our experimental setup, we explore the performance of `SnapKV` across models that can handle extended prompt sequence contexts. First, we deliver a pressure test and benchmark the speed of `LWM-Text-Chat-1M` [17], which is state-of-the-art regarding its context length. We then conduct an ablation study on `Mistral-7B-Instruct-v0.2` to understand the influence of pooling on the model's information retrieval performance. We assess model performances using the LongBench [18] dataset. Further experiments on compatibility with other acceleration strategies, such as parallel decoding [19], are elaborated in Appendix A. To assess the overhead of `SnapKV` during the prefilling stage, we present time and memory analysis results in Appendix B.

## 5.1   Benchmarks on LWM-Text-Chat-1M

`LWM-Text-Chat-1M` [17] is a 7B instruction-fine-tuned model with up to one million context length. In this section, we conduct a pressure test on this model and examine its algorithmic efficiencies.

### 5.1.1   Needle-in-a-Haystack

The Needle-in-a-Haystack test [20] challenges the model to accurately retrieve information from a specific sentence ("needle") concealed within an extensive document (the "haystack"), with the sentence placed at a random location. Typically, sentences that are inserted in the middle of prompts are harder to retrieve. To rigorously evaluate `SnapKV`'s capabilities, we extended the document length to 380k tokens which is the longest content that can be processed by a single A100-80GB GPU. We configured the prompt KV cache size to 1024, enabling `SnapKV` to select the most crucial 1024 attention features from the prompt for answer generation, with a maximum pooling kernel size of 5 and an observation window size of 16, both of which are hyperparameters that can be customized. The compelling outcomes in Fig. 6 from the Needle-in-a-Haystack test underscore `SnapKV`'s potential to precisely manage small details on extremely long input contexts with a 380x compression ratio.

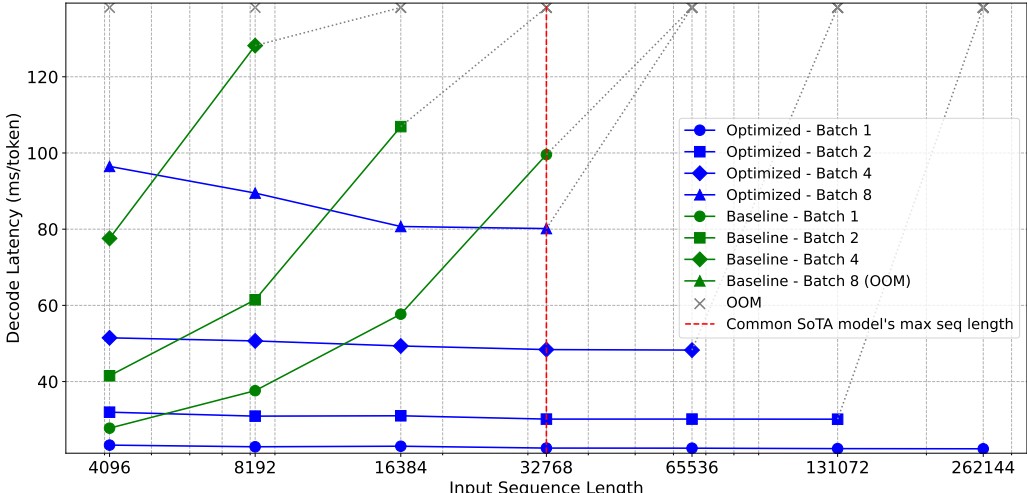

Figure 7: Decoding latency comparison of baseline implementation and `SnapKV` optimized solutions on various batch sizes. The x-axis denotes the input sequence length; the y-axis indicates decoding latency (ms/token). All experiments are conducted on an A100 80GB GPU. The red dotted line denotes the common context length of state-of-the-art long sequence models.

### 5.1.2 Decoding Speed and Memory Bound

We further benchmark the speed of `LWM-Text-Chat-1M` under different batch-size settings using `SnapKV`. We set the maximum KV cache size as 2048 for `SnapKV`, and fix the generation length at 512 to ensure a fair comparison. There are two main takeaways from our experiment on decoding speed and prompt sequence length on various batch sizes, as shown in Fig. 7. First, as the input sequence length increases, the decoding latency of the baseline implementation escalates linearly. Conversely, the `SnapKV`-optimized model maintains a constant decoding speed since the compressed KV cache size of prompt stays the same regardless of input sequence length and there is no extra update during the inference. For instance, at a sequence length of 16k and a batch size of 2, the decoding time for the baseline model surpasses 100 ms, whereas for `SnapKV`-optimized model, the decoding time consistently remains below 40 ms, achieving approximately a 3.6x speedup. Second, with the same batch size, the model integrated with `SnapKV` can decode significantly longer sequences. For example, at a batch size of 2, the baseline model encounters an OOM error beyond 16k input tokens, whereas the `SnapKV`-enhanced model extends this limit to 131k input tokens, indicating an approximately 8.2x improvement. This demonstrates `SnapKV`'s effectiveness in minimizing memory consumption.

### 5.2 Ablation Study of Effectiveness of Pooling

We perform an ablation study on `Mistral-7B-Instruct-v0.2` to assess the impact of our pooling technique, a straightforward but efficient method for consolidating information through clustering. Our evaluation utilizes the modified LongEval-Lines benchmark [21], incorporating randomly generated pairs and averaged scores. LongEval-Lines presents a greater challenge compared to Needle-in-a-Haystack because it involves identifying key-value pairs in noisy contexts of the same format, while in Needle-in-a-Haystack, the relevant information is more distinctly separated from other contexts. We apply max pooling with a kernel size of 5 and use the observation window with a size of 16, which are hyperparameters and could be customized according to different models. As illustrated in our results (Fig. 8), we find that pooling significantly enhances retrieval accuracy compared to methods not utilizing pooling. We hypothesize that this is because the initial portions of critical token clusters are weighted higher by attention mechanisms. Typically, large language models tend to copy the tokens surrounding the initial portions to keep the contextual integrity. However, naively compressed KV cache breaks this mechanism and could lead to partially correct

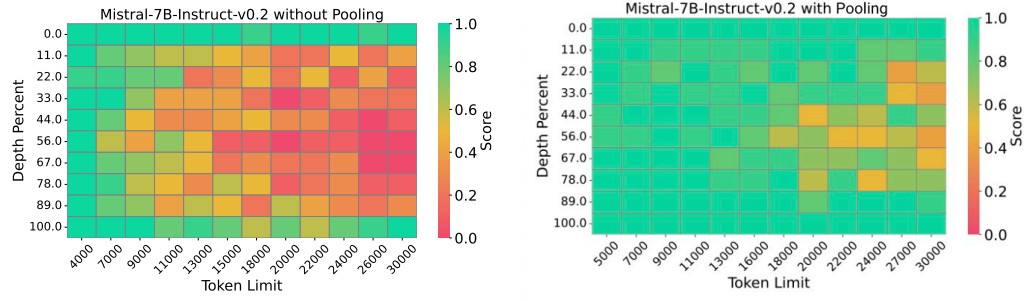

Figure 8: Ablation study of pooling on LongEval-Lines. The evaluation includes inputs, each comprised of lines formatted as `"line makeshift-penguin: REGISTER_CONTENT is <10536>"`, where the key is an adjective-noun pair and the value is a random 5-digit number. The model needs to retrieve the value based on a given key. The x-axis denotes the length of the input; the y-axis indicates the position of the groundtruth, from 5K to 30K tokens. With the pooling, the model can retrieve correct values before 16k and performs significantly better than the one without pooling.

Table 1: Performance comparison of `SnapKV` and H2O across various LLMs on LongBench.

| | LLMs * | Single-Document QA | | | Multi-Document QA | | | Summarization | | | Few-shot Learning | | | Synthetic | | Code | |
| | | NrtvQA | Qasper | MF-en | HotpotQA | 2WikiMQA | Musique | GovReport | QMSum | MultiNews | TREC | TriviaQA | SAMSum | PCount | PRe | Lcc | RB-P |
|---|---|---|---|---|---|---|---|---|---|---|---|---|---|---|---|---|---|
| **LWMChat** | All KV | **18.18** | **25.56** | 40.94 | 24.57 | 19.39 | 10.49 | **27.97** | 24.9 | **24.81** | 71.0 | 60.9 | 39.73 | 3.17 | 3.5 | 44.4 | 43.82 |
| | SnapKV: 1024 | 18.02 | 23.73 | 40.25 | 24.61 | **19.84** | 10.77 | 19.79 | 24.44 | 23.53 | 70.0 | **61.42** | 39.64 | 1.67 | 3.0 | 43.34 | 44.0 |
| | SnapKV: 2048 | 17.92 | 25.03 | **41.38** | 24.49 | 19.38 | **11.34** | 21.6 | 24.22 | 24.36 | 70.0 | 61.11 | 39.91 | 2.17 | 4.0 | 44.46 | **44.92** |
| | SnapKV: 4096 | 17.92 | 25.47 | 40.76 | **24.92** | 19.53 | 11.27 | 25.34 | **25.42** | 24.58 | 70.5 | 61.08 | 39.62 | **3.17** | **4.0** | **44.49** | 44.08 |
| | H2O: 4096 | 13.17 | 24.82 | 20.01 | 16.86 | 9.74 | 7.2 | 25.77 | 23.26 | 23.83 | **71.0** | 61.06 | **40.33** | 0.0 | 0.0 | 41.52 | 40.97 |
| **LongChat** | All KV | **20.88** | **29.36** | **43.2** | 33.05 | 24.58 | **14.66** | **30.89** | 22.76 | **26.61** | 66.5 | **83.99** | **40.83** | 0.0 | 30.5 | 54.89 | **59.05** |
| | SnapKV: 1024 | 19.32 | 26.6 | 37.93 | 34.15 | 23.34 | 12.71 | 23.45 | 21.81 | 24.93 | 65.0 | 80.88 | 38.19 | 0.0 | 31.0 | 53.63 | 57.62 |
| | SnapKV: 2048 | 19.28 | 28.81 | 40.26 | **35.31** | 23.75 | 13.44 | 26.3 | 22.29 | 25.73 | 66.0 | 79.93 | 39.59 | 0.0 | **31.0** | **56.05** | 58.61 |
| | SnapKV: 4096 | 20.68 | 29.34 | 42.21 | 33.95 | **24.88** | 14.15 | 28.55 | **23.11** | 26.45 | 66.0 | 81.25 | 40.52 | 0.0 | 29.5 | 54.79 | 58.81 |
| | H2O: 4096 | 19.31 | 28.3 | 37.75 | 30.51 | 23.06 | 11.76 | 27.55 | 21.37 | 26.49 | 66.0 | 75.8 | 39.92 | 0.0 | 25.5 | 53.56 | 55.53 |
| **Mistral** | All KV | **26.82** | 33.06 | 49.28 | **42.77** | 27.33 | 19.27 | **32.85** | 24.25 | 27.06 | 71.0 | 86.23 | 42.98 | 2.75 | 86.98 | 55.51 | **52.88** |
| | SnapKV: 1024 | 25.54 | 29.51 | 49.25 | 40.94 | 25.7 | **19.42** | 25.89 | 23.82 | 26.11 | 69.5 | **86.48** | 42.06 | 2.98 | **88.56** | 55.65 | 51.87 |
| | SnapKV: 2048 | 25.89 | 32.47 | 48.6 | 41.71 | 27.31 | 18.69 | 28.81 | **24.5** | 26.6 | 70.0 | 86.27 | 42.47 | 3.09 | 87.43 | **55.93** | 52.01 |
| | SnapKV: 4096 | 26.41 | **33.36** | **49.81** | 42.32 | **27.93** | 18.76 | 30.74 | 24.19 | **27.08** | **71.0** | 86.25 | **43.01** | 2.73 | 86.18 | 55.62 | 52.65 |
| | H2O: 4096 | 22.61 | 29.06 | 47.22 | 36.54 | 20.6 | 16.25 | 30.0 | 23.8 | 26.75 | 70.5 | 86.16 | 42.97 | **3.46** | 86.38 | 53.72 | 51.1 |
| **Mixtral** | All KV | 26.81 | **37.06** | 51.55 | 47.77 | 32.46 | **26.59** | 34.25 | **26.05** | 27.91 | 76.0 | 90.57 | 46.98 | 5.5 | 100.0 | **69.07** | 69.65 |
| | SnapKV: 1024 | 26.01 | 34.65 | 51.58 | **48.23** | 32.67 | 25.92 | 27.77 | 25.0 | 27.25 | 74.5 | 90.42 | 46.48 | 5.5 | 99.5 | 69.02 | 68.98 |
| | SnapKV: 2048 | **27.12** | 36.9 | 51.91 | 47.46 | 33.23 | 26.27 | 30.19 | 25.84 | 27.8 | **76.0** | 90.24 | 46.31 | 5.5 | 100.0 | 68.72 | **70.01** |
| | SnapKV: 4096 | 26.46 | 37.03 | **52.62** | 47.71 | **33.35** | 26.45 | 32.64 | 25.87 | **27.94** | 75.5 | **90.71** | **47.14** | 5.5 | **100.0** | 68.81 | 69.56 |
| | H2O: 4096 | 20.45 | 32.09 | 48.02 | 34.76 | 25.69 | 16.5 | 29.76 | 23.53 | 26.84 | 74.5 | 90.24 | 47.1 | **7.06** | 99.42 | 64.91 | 63.52 |

\* Credit to Jin et al. [22] for the template used in the table.

results (Fig. 8). Note that throughout our experiments, the choice between max pooling and average pooling did not yield significant differences in performance.

## 5.3 Experiments on LongBench

We evaluate `SnapKV` on these four models using LongBench [18], a multi-task benchmark designed to rigorously evaluate long context understanding capabilities across various datasets, spanning single and multi-document QA, summarization, few-shot learning, synthetic tasks, and code completion. The average prompt length of LongBench ranges from 5k to 7k, and more details can be found in Appendix D. We choose `LWM-Text-Chat-1M` with 1 million context length, `LongChat-7b-v1.5-32k`, `Mistral-7B-Instruct-v0.2`, `Mixtral-8x7B-Instruct-v0.1` with 32k context length as our baselines. For each model, we test `SnapKV` with various settings: compressing KV caches in the prompt to 1024, 2048, and 4096 tokens. We use max pooling with kernel size 7 and observation window size 32. Table 1 illustrates a negligible performance drop from models with `SnapKV` compared with original implementations for 16 different datasets, even with prompt-KV with 1024 tokens. Some models even outperform the baseline. Our results substantiate that `SnapKV` can grasp the key information in the long context and give comprehensive summaries with details. Moreover, our

Table 2: The sensitivity analysis was conducted on `Mistral-7B-Instruct-v0.2` with a prompt KV cache size set to 1024, evaluating its performance on LongBench across different observation window sizes and pooling kernel dimensions. In previous experiments, a configuration with an observation window size w=32 and a kernel size k=7 was employed as the baseline. w=32 k=1 refers to SnapKV without pooling, where we focus on tasks that do not involve information retrieval since we already demonstrate that in Sec. 5.2.

| | | Single-Doc. QA | | Multi-Doc. QA | | Summarization | | Few-shot Learning | | | Synthetic | | Code | |
|---|---|---|---|---|---|---|---|---|---|---|---|---|---|---|
| | LLMs * | Qasper | MF-en | HopotQA | 2WikiMQA | GovReport | MultiNews | TREC | TriviaQA | SAMSum | PCount | PRe | Lcc | RB-P |
| Mistral-7B | w=32 k=7 | **29.51** | **49.25** | 40.94 | 25.7 | 25.89 | 26.11 | **69.5** | 86.48 | **42.06** | 2.98 | 88.56 | 55.65 | **51.87** |
| | w=16 k=7 | 27.14 | 48.9 | 41.02 | **27.06** | 28.2 | 26.13 | 67.0 | 86.84 | 40.9 | 4.51 | **91.56** | 60.55 | 50.25 |
| | w=64 k=7 | 27.28 | 48.99 | 40.95 | 26.95 | 26.41 | **26.18** | 67.0 | 86.84 | 40.85 | 4.44 | **91.56** | **60.79** | 50.25 |
| | w=32 k=5 | 26.79 | 48.7 | 40.07 | 26.74 | 29.65 | 24.55 | 64.29 | 86.73 | 40.21 | **4.74** | 90.49 | 57.06 | 48.57 |
| | w=32 k=9 | 27.18 | 49.19 | **41.39** | 26.55 | 26.58 | 24.61 | 65.33 | **86.87** | 39.74 | 4.51 | **91.56** | 60.56 | 50.25 |
| | w=32 k=1 | - | - | - | - | **33.23** | 26.04 | 67.33 | 86.84 | 40.9 | 4.51 | **91.56** | 60.66 | 50.25 |

results also indicate the effectiveness of SnapKV in compressing the prompt KV cache. For these 4 models, the average input token length is around 13k. Thus, using 1024, SnapKV achieves an average compression rate of 92%, and using 4096, it reaches 68%, all with negligible drops in accuracy. We compare SnapKV and H2O on the LongBench dataset to further demonstrate the performance of SnapKV. To fairly evaluate the accuracy, we set the prompt capacity for H2O to 4096. As Table 1 shows, SnapKV delivers significantly better performance than H2O. Even with 1024 prompt KV caches, SnapKV on `Mistral-7B-Instruct-v0.2` achieves better performance than H2O with 4096 caches on 11 out of 16 benchmarks.

## 5.4 Sensitivity Analysis on Hyperparameters

In SnapKV, we introduce two key hyperparameters: observation window size and pooling kernel size. To further assess the robustness of our method, we perform a sensitivity analysis on `Mistral-7B-Instruct-v0.2` with these hyperparameters using the LongBench dataset. As shown in Table 2, different configurations yield the best score across various types of tasks, with no single configuration consistently outperforming others. This indicates that SnapKV demonstrates robustness across a range of configurations.

Additionally, to better understand the effectiveness of the pooling strategy, we conduct an experiment with a kernel size of 1, representing a configuration without pooling. This analysis focuses primarily on non-retrieval tasks, complementing the retrieval task results in Sec. 5.2. The findings indicate that, in eight out of nine tasks, the model accuracy with pooling exceeds that of configurations without pooling, underscoring the importance of pooling in SnapKV.

## 6 Discussions

SnapKV is an effective yet straightforward solution that compresses the KV cache to mitigate the computational and memory burdens of processing extensive prompts. Observing that specific tokens within prompts gain consistent attention from each head during generation, our methodology not only retrieve crucial information but also enhances processing efficiency. Despite its strengths, SnapKV's scope is primarily confined to the generative aspect of models, specifically targeting the KV cache during the generation. This limitation implies that SnapKV cannot extend a model's long context capability if the model inherently struggles with long contexts or exhibits poor performance. Additionally, SnapKV's design does not cover the processing of the prompt inference, which limits its effectiveness in scenarios where the system cannot handle prompts of extensive length. Nonetheless, our contributions offer significant insights and tools for the community, paving the way for more refined approaches on managing the challenges of large-scale language modeling. The appendix provides more experiments with parallel decoding and the discussion about generation speedup.

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

# A    Case Study: Compatibility with Parallel Decoding

In this section, we provide a novel perspective on employing KV cache compression synergistically with parallel decoding [23–27]. Parallel decoding leverages a lightweight model or an adaptor to draft initial tokens, which are subsequently verified by larger LLMs. This strategy effectively reduces memory overhead, a critical concern given the autoregressive nature of LLMs that renders them more memory-intensive than computationally demanding. Specifically, in LLMs, each decoding step involves generating a single token, with the transfer of weights between High Bandwidth Memory (HBM) and cache contributing to significant overhead [28, 29].

Our investigation incorporates `SnapKV` with `Medusa` [19][3], a cutting-edge parallel decoding framework that utilizes multiple classifiers and tree attention mechanisms for drafting tokens, subsequently verified by LLMs. One of the challenges identified is the issue of speculative decoding in processing long sequences since generating multiple tokens per decoding step introduces computational bottlenecks during long sequence processing, such as query-key matrix multiplication tiling [30]. By maintaining a constant size for the KV cache associated with prompts during generation, `SnapKV` enhances generation efficiency.

Empirical results shown in Figure 9 highlight the performance across various prompt lengths, with `Mistral-7B-Instruct-v0.2`[4] undergoing a maximum of 128 generation steps unless preemptively halted. The experiments utilized a subset of the QASPER [31], with a fixed prompt instructing the LLM to summarize the paper. The truncation strategy adopted aligns with LongBench [18] standards, by removing the context in the middle to achieve the desired sequence length for benchmarking.

The findings indicate a slowdown in `Medusa`'s performance as sequence lengths extend, a challenge effectively mitigated by `SnapKV`'s intervention, which achieved a 1.3x speedup for sequences with 10k length compared to `Medusa` and a 2.2x speedup compared to the native decoding. This improvement underscores the potential of combining KV cache compression with parallel decoding frameworks to enhance LLM efficiency, particularly in long-context scenarios.

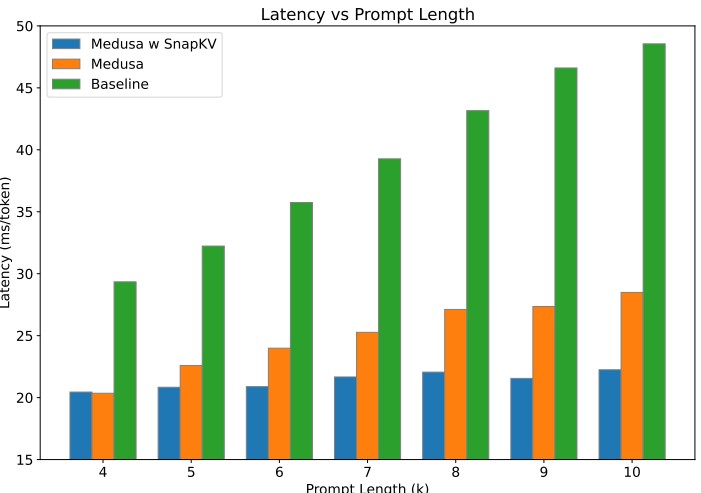

Figure 9: Comparison of generation latency (ms/token). The baseline is the Huggingface implementation of naive decoding.

---

[3]`https://github.com/FasterDecoding/Medusa`
[4]TGI trained `Medusa` heads

# B  Overhead Analysis of Prefilling Stage

We evaluate the prefilling time and memory usage on `Mistral-7B-Instruct-v0.2` with input sequence lengths ranging from 5k to 45k in Fig. 10. The results show no overhead in either aspect. SnapKV only introduces extra top-k and pooling operations which are trivial regarding computation complexity compared with original prefilling calculations.

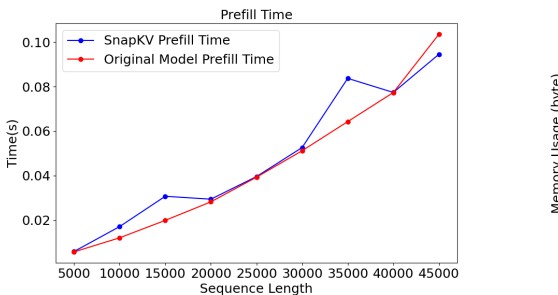 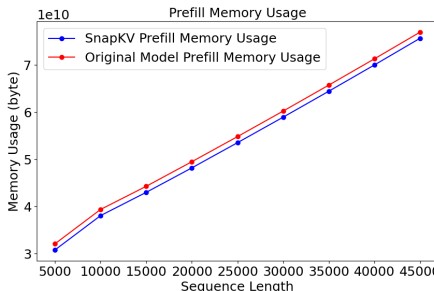

Figure 10: The prefilling time and maximum memory allocated comparison between `Mistral-7B-Instruct-v0.2` with and without SnapKV on an H100.

# C  Discussion of Generation Time Speedup

To better assess `SnapKV`'s effectiveness across different stages, we documented a detailed time breakdown for `Mistral-7B-Instruct-v0.2` during both the prompting and generation stages. We configured the model to consistently generate 512 tokens, facilitating a direct comparison with the prompting stage. As illustrated in Fig. 11, generation time dominates the whole processing time for LLMs over input sequences, introducing significant overhead. While the generation time for the original model increases with input length, `SnapKV` maintains a consistent decoding speed regardless of input length, significantly reducing generation time. Especially, `SnapKV` is able to achieve balanced prompting time and generation time with input length smaller than 100k.

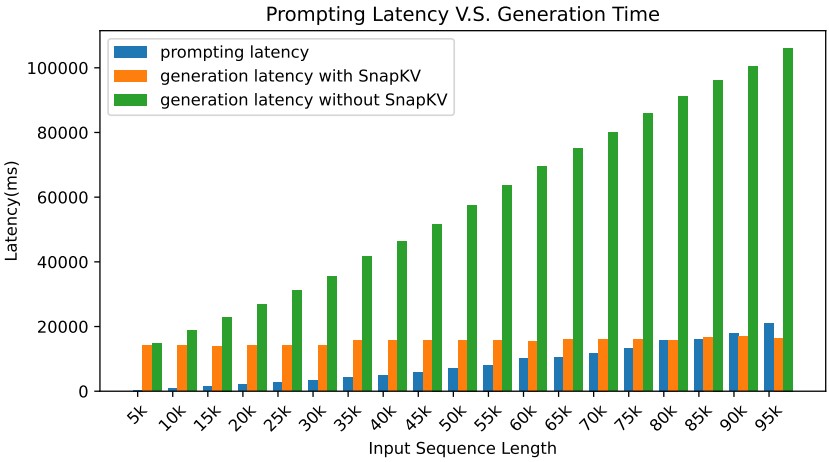

Figure 11: The prompting time and generation time comparison between Mistral model with and without SnapKV.

# D   Statistics of LongBench Dataset

Here we provide more statistical information on LongBench in Table 3 to provide more insights on our results in 5.3.

Table 3: Sequence Length Statistics for LongBench Benchmark

| Dataset | Min | Max | Avg |
|---|---|---|---|
| Qasper | 1443 | 14722 | 4620 |
| HotpotQA | 111 | 12480 | 6658 |
| 2WikimQA | 881 | 23442 | 7141 |
| GovReport | 111 | 12480 | 6658 |
| MultiNews | 374 | 27973 | 6000 |
| MF-en | 505 | 10337 | 4559 |
| TREC | 746 | 13034 | 5475 |
| TriviaQA | 804 | 15960 | 6685 |
| SAMSum | 936 | 12403 | 6170 |
| PCount | 1407 | 14537 | 6117 |
| PRe | 2358 | 10607 | 6115 |
| Lcc | 386 | 14106 | 4283 |
| RB-P | 785 | 18864 | 6067 |

# E Visulization of the Generated Context

**Prompt(length 20581) from Qasper**

You are given a scientific article and a question. Answer the question as concisely as you can, using a single phrase or sentence if possible. If the question cannot be answered based on the information in the article, write "unanswerable". If the question is a yes/no question, answer "yes", "no", or "unanswerable". Do not provide any explanation.
[ Characterizing Political Fake News in Twitter by its Meta-DataJulio Amador Díaz LópezAxel Oehmichen Miguel Molina-Solana( j.amador, axelfrancois.oehmichen11, mmolinas@imperial.ac.uk ) Imperial College London This article presents a preliminary approach towards characterizing political fake news on Twitter through the analysis of their meta-data. In particular, we focus on more than 1.5M tweets collected on the day of the election of Donald Trump as 45th president of the United States of America. We use the meta-data embedded within those tweets in order to look for differences between tweets containing fake news and tweets not containing them.

......

Question: How is the ground truth for fake news established?

Answer:

**Ground Truth**

Ground truth is not established in the paper

**SnapKV: 1024**

The ground truth for fake news is not claimed in the article.

**SnapKV: 2048**

The ground truth for fake news is not claimed in the article as the categorization is far from perfect due to the ambiguity of fake news and human judgement involved in the process.

**SnapKV: 4096**

The dataset is not claimed to be a ground truth due to the ambiguity of fake news and human judgement involved in the categorization process.

**All KV**

The dataset is not claimed to be a ground truth due to the ambiguity of fake news and human judgement involved in the categorization process.

**Prompt(length 31500) from HotpotQA**

Answer the question based on the given passages. Only give me the answer and do not output any other words.

The following are given passages.
Passage 1:
List of Unbreakable Kimmy Schmidt episodes
Unbreakable Kimmy Schmidt is an American sitcom created by Tina Fey and Robert Carlock, starring Ellie Kemper in the title role, that has streamed on Netflix since March 6, 2015. Originally set for a 13-episode first season on NBC for spring 2015, the show was sold to Netflix and given a two-season order. The fourth and final season concluded on January 25, 2019. On May 8, 2019, it was announced that the series would return with an interactive special, which premiered on May 12, 2020. During the course of the series, 52 episodes of Unbreakable Kimmy Schmidt were released over four seasons.

......

Answer the question based on the given passages. Only give me the answer and do not output any other words.

Question: Who stars in the title role of a Netflix show co-created by Robert Carlock?
Answer:

**Ground Truth**

Ellie Kemper

**SnapKV: 1024**

Ellie Kemper stars in the title role of Unbreakable Kimmy Schmidt.

**SnapKV: 2048**

Ellie Kemper stars in the title role of Unbreakable Kimmy Schmidt.

**SnapKV: 4096**

Ellie Kemper stars in the title role of Unbreakable Kimmy Schmidt.

**All KV**

Ellie Kemper stars in the title role of Unbreakable Kimmy Schmidt.

Figure 12: Visualization of generation examples from Samsum, Qasper, HotpotQA datasets with `mistral-7B-instruct-v0.2`. Results are compared between ground truth, `SnapKV` with 1024 prompt tokens, with 2048, with 4096, the baseline model with full KV cache.

