# OpenReview forum: "SnapKV: LLM Knows What You are Looking for Before Generation"
_NeurIPS.cc/2024/Conference — NeurIPS 2024 poster_

### Official Review · Reviewer_Akz1 · 2024-07-10

**Soundness:** 2
**Presentation:** 2
**Contribution:** 2
**Rating:** 6
**Confidence:** 4

**Summary:**

This paper analyzes the issue of high storage pressure in KV caches within long-context scenarios and proposes a method for KV cache compression. Specifically, it calculates the attention weights between the last 16 Q tokens and K, and utilizes pooling and topK to determine the KV pairs to retain. The paper tests four models—LWM-1M, Mistral-32K, LongChat-32K, and Mixtral-32K—on two tasks, LongBench and Needle In A Haystack. The results show a 2% drop on the GovReport task and a reduction to 140K context on the Needle In A Haystack task, while in other tasks, the results are generally consistent with full attention. The decoding stage latency is reduced by 3.6x.

**Strengths:**

- The problem studied in the paper is significant and has practical value.
- The motivation of the paper is sound, supported by ample experimental evidence.

**Weaknesses:**

1. The experimental section is limited, with tests conducted only on two benchmarks, making it difficult to demonstrate the method's generalizability and effectiveness. For the Needle In a Haystack task, only the results of LWM-1M with SnapKV are tested, lacking comparison with full attention, other baselines, and results from other models. Additionally, the tested context windows are relatively short, only up to 380K.
2. The paper uses a pooling method in the approximate stage and claims it is for more efficient pooling. However, Fig. 8 shows significant performance improvements with pooling, leading to doubts about whether the important Keys identified by the observation Query remain unchanged during the decoding stage, especially in retrieval tasks or tasks like Needle in the haystack.

**Questions:**

1. Do you have additional baseline results for the Needle In A Haystack test, such as those for H20 or the full-attention model?
2. Are there any results from more demanding long-context benchmarks, such as RULER[1] or InfiniteBench[2]? I am particularly curious about how SnapKV would perform on tasks akin to KV retrieval[2]. Furthermore, do you have experimental results from other, more capable long-context language models, such as Yi-200K[3] or LLaMA-3-1M[4]?
3. Have you conducted experiments on contexts longer than those reported, and if so, could you share those results?
4. Regarding the speedup data in Sec 5.1.2, are these results based on the vLLM freamwork? If not, could you provide results for vLLM, as they would more accurately reflect the degree of speedup improvement in real-world scenarios?
5. Typo: In Fig. 8, the right image's label "without" should be corrected to "with."

- [1] RULER: What’s the Real Context Size of Your Long-Context Language Models?
- [2] InfiniteBench: Extending Long Context Evaluation Beyond 100K Tokens
- [3] https://huggingface.co/01-ai/Yi-34B-200K
- [4] https://huggingface.co/gradientai/Llama-3-8B-Instruct-Gradient-1048k

**Limitations:**

Please refer to the weaknesses section.

---

> ### Author Rebuttal · Authors · 2024-08-06
>
> *Q1. The paper uses a pooling method in the approximate stage and claims it is for more efficient pooling. However, Fig. 8 shows significant performance improvements with pooling, leading to doubts about whether the important Keys identified by the observation Query remain unchanged during the decoding stage, especially in retrieval tasks or tasks like Needle in the haystack.*
>
> **Answer:** Our observation reflects the consistent pattern during the decoding stage. The reason behind the effectiveness of pooling is that it could keep the integrity of information. For example, there is a number 123456, and SnapKV finds 34 are the important tokens. If we only retrieve 34, it cannot keep the original number, but with pooling, we can select 12 and 45 as well to retain the whole information.
>
> *Q2. Are there any results from more demanding long-context benchmarks, such as RULER[1] or InfiniteBench[2]? I am particularly curious about how SnapKV would perform on tasks akin to KV retrieval[2]. Do you have experimental results from other, more capable long-context language models, such as Yi-200K[3] or LLaMA-3-1M[4]?*
>
> **Answer:** Thanks for the valuable feedback. Because of the time limit, we are unable to evaluate these benchmarks, but we evaluate on Llama-3-1M as you referred. We evaluate one dataset for every category in the LongBench benchmark in PDF Table 2. The results suggest that SnapKV (compressed to 1024 tokens with the observation window equal to 32 and pooling kernel equal to 7) can be adapted to the Llama model and match or even outperform the original model on various tasks.
>
> *Q3. Do you have additional baseline results for the Needle In A Haystack test, such as those for H20 or the full-attention model? Have you conducted experiments on contexts longer than those reported, and if so, could you share those results?*
>
> **Answer:** For the Needle In a Haystack task, the original LWM model will encounter an OOM error after 30k sequence length. Additionally, we test the Needle In a Haystack on H2O.  However, H2O needs to compute the attention weight of every decoding step during the generation to decide which KV cache to drop. This mechanism of H2O makes it incompatible with FlashAttention (aims to not store full attention weight), which results in even faster OOM error (less than 30k) on a single A100. This result also reveals that H2O cannot solve the long context problem in general. In addition, all our tests are based on a single A100 with 80G GPU, which limits the sequence length to 380K.
>
> *Q4. Regarding the speedup data in Sec 5.1.2, are these results based on the vLLM framework? If not, could you provide results for vLLM, as they would more accurately reflect the degree of speedup improvement in real-world scenarios?*
>
> **Answer:** Thanks for the suggestion. Because LLM serving frameworks like vLLM incorporate many optimizations on parallelization, memory management, etc., it is unfair for end-to-end comparison. [Link][1] is an independent study across different LLM serving frameworks including vLLM, TensorRT, etc. It shows the benefit of this framework on input sequence lengths ranging from 20 to 5000. As SnapKV typically compresses KV cache to 1024 to 4096, we expect the LLM serving framework and SnapKV can mutually benefit from each other, and achieve stacked speedup.
>
> [1]: https://www.inferless.com/learn/exploring-llms-speed-benchmarks-independent-analysis

---

> > ### Comment · Reviewer_Akz1 · 2024-08-11
> > **Official Comment by Reviewer Akz1**
> >
> > Thank you for your detailed response. I understand the motivation behind the pooling method; however, I remain skeptical about the fundamental reasons for the gains attributed to pooling. It would be beneficial to include more analysis in future versions. Overall, SnapKV represents an excellent piece of work in KV cache compression, with clear motivation and significant application potential. I have raised my score to 6.

---

> > > ### Author Response · Authors · 2024-08-13
> > >
> > > Thanks for your valuable feedback! We will continue refining our paper in the future version.

---

### Official Review · Reviewer_wNH7 · 2024-07-12

**Soundness:** 3
**Presentation:** 3
**Contribution:** 3
**Rating:** 7
**Confidence:** 5

**Summary:**

This paper presents a training-free KV cache compression approach. This approach builds off of the observation that the model consistently focuses on particular features during generation, and that these features can be detected when the prompt is passed in. Their approach uses a small “observation window” of queries at the end of the prompt to detect which previous keys must be retained, and then prunes out other key / value pairs. They also incorporate a pooling-based approach in order to maintain contextual information around important tokens. Through these methods, they attain generation speedups and reduced memory consumption for long context length tasks, while maintaining accuracy.

**Strengths:**

- Their paper provides multiple important insights about how models use their input context; namely, the insight that the important tokens during the prompt processing phase stay consistent throughout the generation process, as well as the observation that different instructions prioritize different parts of the input prompt (which therefore necessitates a dynamic approach)
- Their approach compresses the KV cache for the input prompt, which is crucial for many long context length applications (where the majority of the context length is used up by a long prompt), while maintaining accuracy
- Their approach is relatively simple and is therefore compatible with existing kernel methods (and can be implemented using only Pytorch-level code changes)
- Significant accuracy improvements over prior work (H2O) when processing long context length tasks

**Weaknesses:**

- This paper doesn’t accelerate the prompt processing step (even though it is explicitly targeting long prompt processing), which could present a bottleneck for tasks with long input lengths and short generation lengths
- The paper lacks benchmarking experiments to justify that their approach doesn’t add any overheads to the prompt processing step (due to having to use the observation window to identify important tokens, which likely needs to be applied separately from FlashAttention)
- They do not provide sufficient justification for their pooling approach, and their ablations are insufficient to show its effectiveness (they show accuracy benefits on one task, but for other tasks with less sparse attention distributions than retrieval this may actually degrade performance by prioritizing less important tokens that are near important ones). It would be good to ablate the impacts of pooling on non-retrieval tasks as well (eg. few-shot ICL) which may have a flatter attention distribution, and where pooling may lead to retrieving less of the actual important information that is required.

**Questions:**

- For Table 1, it would be good to also include the average input prompt sizes for the different tasks (in order to get a sense of the relative compression that is attained by going to fixed cache sizes of 1K, 2K, 4K)

**Limitations:**

Yes

---

> ### Author Rebuttal · Authors · 2024-08-06
>
> *Q1. The paper lacks benchmarking experiments to justify that their approach doesn’t add any overheads to the prompt processing step.*
>
> **Answer:** Thanks for the valuable feedback. To address your comment, we evaluate the prefilling time and memory usage on Mistral-7B with input sequence lengths ranging from 5k to 45k in PDF Figure 1 & 2. The results show no overhead in either aspect. SnapKV only introduces extra topk and pooling operations which are trivial regarding computation complexity compared with original prefilling calculations.
>
> *Q2. They do not provide sufficient justification for their pooling approach, and their ablations are insufficient to show its effectiveness (they show accuracy benefits on one task, but for other tasks with less sparse attention distributions than retrieval, this may degrade performance by prioritizing less important tokens that are near important ones). It would be good to ablate the impacts of pooling on non-retrieval tasks as well (eg. few-shot ICL) which may have a flatter attention distribution, and where pooling may lead to retrieving less of the actual important information that is required.*
>
> **Answer:** We conduct ablation experiments on the effectiveness of pooling on the LongBench dataset and the results are presented in PDF Table 1 as w=32 k=1. In eight out of nine tasks, the model accuracy with pooling is better than those without pooling.
>
> *Q3. For Table 1, it would be good to also include the average input prompt sizes for the different tasks.*
>
> **Answer:** Thanks for the feedback, we provide more information about the LongBench benchmark in PDF Table 3. In general, the sequence lengths approximately range from 100 to 23000 with an average of 5817.

---

> > ### Comment · Reviewer_wNH7 · 2024-08-08
> > **Rebuttal Response**
> >
> > Thank you for your work in your rebuttal.
> > - For the first question, the author's response clearly demonstrates that their approach doesn't add runtime overhead in the prefill phase.
> > - I also appreciate the inclusion of the average prompt sizes for each task, as this helps understand the compression achieved in each case.
> > - For the pooling ablation (Q2), the data does not seem as clear as suggested that the pooling approach yields consistent benefits. When compared with the configuration used in the paper, the average accuracy is actually higher for the configuration without pooling, and it does not seem like any of the configurations are consistently superior. I still feel that the benefits of pooling across a diverse range of tasks are unclear.

---

> > > ### Author Response · Authors · 2024-08-13
> > >
> > > Thanks for your valuable feedback. From the experiments, SnapKV with pooling performs better than without it in most cases, but indeed, various tasks require different configurations, including observation windows and pooling kernel sizes. We will continue refining our work and discuss this situation in future versions.

---

### Official Review · Reviewer_Zgi7 · 2024-07-13

**Soundness:** 3
**Presentation:** 3
**Contribution:** 2
**Rating:** 6
**Confidence:** 3

**Summary:**

The paper introduces a method for minimizing KV cache size in LLMs. The authors offer the insight that attention heads focus on specific prompt attention features (tokens and their feature representation) during generation. This pattern can be discovered via an observation window at the end of the prompt. The new method, SnapKV, compresses KV caches by selecting clustered important KV positions for each attention head. As a result, the computational overhead and memory footprint are significantly reduced for long inputs. The method is evaluated across various LLMs and datasets showing its efficiency in practical applications.

**Strengths:**

The paper addresses a significant challenge in LLM efficiency for long context processing.
The insight about consistent attention patterns is useful and may generate new ideas.
The results show some impressive performance improvements and ability to handle very long sequences.
The method is fine-tuning free, and can be integrated into existing frameworks.

**Weaknesses:**

The impact on model accuracy for very long context (more than 100K tokens) is not thoroughly explored.

**Questions:**

How sensitive is the method to the observation window size, or the pooling kernel size?
It would be useful to include some more details about the performance measures used in Table 1.

**Limitations:**

yes

---

> ### Author Rebuttal · Authors · 2024-08-06
>
> *Q1. How sensitive is the method to the observation window size, or the pooling kernel size? It would be useful to include some more details about the performance measures used in Table 1.*
>
> **Answer:** Thanks for the valuable feedback. To address your comment, we experiment with SnapKV on various observation window sizes and pooling kernel sizes. The results can be found in PDF Table 1. Different configuration performs differently in various kinds of datasets as expected.

---

### Author Rebuttal · Authors · 2024-08-06

Thank you for all the questions and suggestions. The PDF file contains all the tables and figures mentioned in the rebuttal.

---

### Decision · Program_Chairs · 2024-09-25

**Decision:**

Accept (poster)

**Comment:**

This paper discussed an important problem in improving the efficiency of long context inference. The solution is solid with good novelty, and the experimental section is well designed to verify the design of the proposed solution.